# Molecular Mechanisms of the Acute Kidney Injury to Chronic Kidney Disease Transition: An Updated View

**DOI:** 10.3390/ijms20194941

**Published:** 2019-10-06

**Authors:** Francesco Guzzi, Luigi Cirillo, Rosa Maria Roperto, Paola Romagnani, Elena Lazzeri

**Affiliations:** 1Excellence Centre for Research, Transfer and High Education for the development of DE NOVO Therapies (DENOTHE), University of Florence, 50139 Florence, Italy; paola.romagnani@unifi.it (P.R.); elena.lazzeri@unifi.it (E.L.); 2Department of Experimental and Clinical Biomedical Sciences “Mario Serio”, University of Florence, 50134 Florence, Italy; 3Nephrology and Dialysis Unit, Meyer Children’s University Hospital, 50139 Florence, Italy; cirlui89@gmail.com (L.C.); rosa.roperto@meyer.it (R.M.R.)

**Keywords:** acute kidney injury, chronic kidney disease, renal progenitors, polyploidization

## Abstract

Increasing evidence has demonstrated the bidirectional link between acute kidney injury (AKI) and chronic kidney disease (CKD) such that, in the clinical setting, the new concept of a unified syndrome has been proposed. The pathophysiological reasons, along with the cellular and molecular mechanisms, behind the ability of a single, acute, apparently self-limiting event to drive chronic kidney disease progression are yet to be explained. This acute injury could promote progression to chronic disease through different pathways involving the endothelium, the inflammatory response and the development of fibrosis. The interplay among endothelial cells, macrophages and other immune cells, pericytes and fibroblasts often converge in the tubular epithelial cells that play a central role. Recent evidence has strengthened this concept by demonstrating that injured tubules respond to acute tubular necrosis through two main mechanisms: The polyploidization of tubular cells and the proliferation of a small population of self-renewing renal progenitors. This alternative pathophysiological interpretation could better characterize functional recovery after AKI.

## 1. AKI Is Not a Self-Limiting Event

Despite the common belief of a generally benign nature, the profound, long-term implications of acute kidney injury (AKI) are appearing more and more evident. In particular, the emergent finding of a progression to various degrees of chronic kidney disease (CKD) after apparently self-limiting AKI episodes, independent from the etiology, has attracted great attention.

In the last decade, the assumption of an existing association between AKI and CKD has gradually spread in the nephrology and intensive care fields with evidence of a tight link between even mild serum creatinine elevation and long-term CKD [1,2,3]. The classic teaching case regarding acute renal failure, in particular acute tubular necrosis (ATN), has been that those patients generally achieve full or nearly full recovery [4,5]. The consensus on a new definition of AKI has helped to improve the understanding of its long-term clinical consequences and to demonstrate a clear link between AKI episodes, their severity, and their outcome [6]. With the help of standardized criteria for the definition of AKI and CKD, diverse observational studies collecting data from large administrative databases have increasingly showed a possible association between these two clinical entities [7,8,9]. Indeed, the more recent assumption is now to consider AKI and CKD as two interconnected syndromes where CKD is a risk factor for AKI and, in the meantime, AKI is a risk factor for the development and progression of CKD [3]. A recent systematic review and meta-analysis by See et al. [7] aimed to quantify the association between AKI and CKD by evaluating the results from 82 studies comprising more than 2 million patients experiencing AKI. The authors confirmed an increase in the risk of new or progressive CKD after AKI (HR 2.67, 95% CI 1.99–3.58) with a gradient of risk across AKI stages, an increased risk of end stage renal disease (ESRD) (HR 4.81, 95% CI 3.04–7.62), and an increased risk of death (HR 1.80, 95% CI 1.61–2.02) [7]. Moreover, previous studies have demonstrated that AKI severity [10], duration [11] and frequency [12] are associated with an increased risk of CKD progression. The AKI–CKD link has also been highlighted in pediatric studies [13,14,15].

Despite continuous progresses in the field and the recent ability to better identify the molecular signature of different renal cell types following acute injury, the mechanisms that drive the transition to chronic disease remain debated [16,17]. The traditional beliefs are now challenged by clinical observations and new advances in experimental transgenic models. Researchers have made significant effort trying to elucidate the pathophysiological link from AKI to CKD on cellular and molecular levels by using experimental models. CKD can occur through several pathologic mechanisms involving one or more of the kidney compartments: Vasculature, the tubule-interstitium or the glomerulus. Microvascular loss occurs along with increased fibrosis, worsening relative hypoxia within the kidney and in particular within the outer medulla. This is associated with changes in pericytes to adopt a pro-fibrotic myofibroblast phenotype. Moreover, consequent to altered oxygen availability, tubular injury and necrosis cause tubular dysfunction, oliguria and reduced glomerular filtration via tubulo-glomerular feedback. Thus, after an ischemic injury, the loss of nephronic mass, with remnant nephron hyperfiltration, renin-angiotensin system (RAS) activation, systemic hypertension and subsequent glomerulosclerosis have been described to pave the way from AKI to CKD [18,19,20]. Regardless of the initial insult, evidence of tubular cell loss and replacement by collagen scars and infiltrating macrophages are associated with further renal functional loss and progression towards end stage renal failure. Experimental models have shown that selective epithelial injury could drive capillary rarefaction, interstitial fibrosis, glomerulosclerosis and progression to CKD, substantiating a direct role for damaged tubular epithelial cells (TECs) [21]. Therefore, tubular epithelial cells have attracted increasing attention [22,23].

A new interpretation of this pathophysiology is that the epithelial tubular cell may allow for a better understanding of this somehow unexpected turn in the AKI natural history. Altogether, there is a need for the further investigation of the AKI-to-CKD transition as a public health priority.

## 2. Pathophysiology of the AKI-to-CKD Transition

From a pathophysiological point of view, microvascular integrity, changes in leukocyte and pericyte behavior, and tubular cell survival and function are all features of both AKI and CKD, and several cellular and molecular pathways have been considered to define the transition process. The main pathological mechanisms which concur to explain the AKI-to-CKD transition include: (i) Endothelial dysfunction, vasoconstriction and vascular congestion [24,25]; (ii) interstitial inflammation and the associated infiltration of monocytes/macrophages, neutrophils, T- and B-cells [26,27,28,29,30]; (iii) fibrosis via myofibroblasts recruitment and matrix deposition [31,32,33]; and (iv) tubular epithelial injury and dysregulated repair [23,34,35] (Figure 1). After a brief description of the main molecular pathways of the endothelial, inflammatory and fibrotic response to injury, we focus on the proximal tubular epithelial cell, the main player of the AKI-to-CKD transition [35].

### 2.1. Endothelial Dysfunction

Capillary rarefaction has been extensively described as a consistent feature of the acutely injured nephron and has been linked to progression as a potential initiator of chronic nephropathy. A reduction in regional renal oxygen delivery leads to inflammation, ischemia, and necrosis, thus reflecting an imbalance between arterial pressure and vascular resistance, with a particularly vulnerable area in the outer stripe of the outer medulla. The existence of a putative bone marrow-derived endothelial progenitor cell (EPC) population has been hypothesized and linked to progression to chronic disease [36]. However, a recent study that combined bone marrow and kidney transplantation in a transgenic murine model demonstrated that no extra-renal cells substantially contribute to endothelial repair after selective injury [37].

Once capillary rarefaction is established, tissue hypoxia, mitochondrial dysfunction, inflammation and subsequent fibrosis occur [24]. The cellular and molecular pathways that underlie capillary rarefaction—in particular, the interplay between hypoxia, anti-angiogenic, and angiogenic factors—have yet to be explained [38]. Among angiogenic factors, vascular endothelial growth factor (VEGF), an endogenous cytokine produced by epithelial cells and directed to endothelial cells, is crucial for the preservation of vascular networks. Its reduced production could promote microvascular dysfunction and morphologic changes in the nephron [39,40]. The effects of its exogenous administration, with restoration of the microvascular density, improved renal blood flow, and reduced fibrogenic activity have been described in swine kidneys [41]. Similar effects, with improved endothelial cell survival and prevention of capillary leakage, resulted from the activation of the endothelium-specific receptor Tie2 through angiopoietin-1 (Ang-1)—a protein produced by vasculature support cells and specialized pericytes [42]. Moreover, the transgenic murine inactivation of endothelial hypoxia-inducible factors (Hif1-α and Hif2-α), as well as the deletion of endothelial sphingosine 1-phosphate receptor 1 (S1-pr1), resulted in increased acute and chronic inflammation and fibrosis after injury without affecting capillary permeability [43,44]. Therefore, there is considerable interest in the potential for Hif-stabilizing agents as therapeutic tools in renal injury [45]. In two models of folic acid-induced AKI and ureteral obstruction, Sirtuin 1 (Sirt 1) inactivation in endothelial cells caused impaired recovery, increased fibrosis, and disease progression [46,47].

Endothelial-to-mesenchymal transition (EndoMT) has been proposed as a contributor to capillary rarefaction, interstitial fibrosis, and, therefore, chronic damage [48]. Indeed, the reduction of EndoMT, obtained by reducing endothelium-specific transforming growth factor β (TGF-β) in a transgenic mouse model, was followed by the preservation of renal blood flow and microvasculature, less tissue hypoxia and tubulointerstitial fibrosis, thereby supporting the hypothesis of a link between EndoMT and chronic changes [49]. Moreover, the renal pericyte is now recognized as a key contributor to vascular stability in response to kidney injury [50].

Pericytes sit in close proximity to the endothelial cells within many organs, where they maintain vascular stability and release factors, including PDGF (platelet-derived growth factor) [51], angiopoietin [52], TGF-β [53], VEGF [54] and sphingosine-1-phosphate [55]. There is now an increasing understanding of the role played by these cells in acute and chronic kidney injury, where they leave their perivascular site in response to injury and differentiate to become myofibroblasts. Thus, either injuries or defects in pericyte function induce their detachment, contributing to both vascular rarefaction and increased fibrosis. In the end, recent advances have elucidated a contribution of endothelial cells and their products to capillary rarefaction, inflammation and tubulogenic pathways in a complex cellular interplay [43,44].

### 2.2. Interstitial Inflammation

Both resident and infiltrating immune cells participate in inflammation, injury and repair in the acute phase of kidney injury; through a tight cross-talk with endothelial cells, epithelial cells, and pericytes, they also contribute to disease progression [56]. The recent identification of a particular subset of renal resident macrophages, located at the abluminal side of the peritubular capillaries and capable of monitoring endothelial transport, has provided a perfect paradigm of the interplay between endothelium and immune system [57]. Resident macrophages seem to form a distinct anatomical and functional unit with the peritubular capillary endothelial cells that have the ability to detect and scavenge small immune complexes, possibly explaining the further recruitment of monocytes and neutrophils, as well as tissue injury in immune complex diseases [57]. This macrophage-endothelial functional unit, with a specific cross-talk at both the cellular and molecular levels, is also likely to be involved in the response to AKI [38].

The link between endothelial cells and inflammation is also suggested by CD169+ monocytes/macrophages that counteract the inflammatory response induced by intercellular adhesion molecule-1 (ICAM-1) expression after ischemic AKI [58]. Because they regulate inflammation, neutrophils infiltration, and because of their paracrine effects on tubular epithelial cells, macrophages can play an important role as determinants of AKI outcomes. Two populations of macrophages have been proposed by in vitro studies on behalf of their chemokine receptor repertoire: Pro-inflammatory or M1-subtype (interleukin (IL)-1, IL-6, and tumor necrosis factor-*α* (TNF-*α*)) and anti-inflammatory or M2-subtype (arginase, mannose receptor, IL-10, and IL-4 receptor-*α*), the first classically activated and the latter alternatively activated [59]. Despite being recently revised in regards to these two populations’ in vivo behavior [60,61], this classification has been useful to identify different responses after acute injury in the kidney and a putative role for these cells in disease progression [28]. Macrophages have contrasting roles in renal injury and repair, first increasing the number of M1-polarized cells and then switching to an M2 phenotype supporting epithelial cell repair. Indeed, the depletion of M2 macrophages in mice with established AKI has resulted in prolongation of renal injury [62].

In a post-ischemic transgenic murine model, IL-1 receptor-associated kinase-M (IRAK-M), specifically expressed by monocyte/macrophages, has been demonstrated to influence the progression of AKI to CKD. IRAK-M expression induces the healing phase by inhibiting the toll-like receptor (TLR) and IL-1 receptor signaling, resolving TNF-α-dependent inflammation, and dampening the M1 pro-inflammatory response, all of which have been found to allow for improved functional recovery and structural regeneration [26]. Macrophages also participate in endogenous repair by secreting cytokines, such as IL-22, and providing ligands for retinoic acid (RA) and Wnt/β-catenin. In an ischemia-reperfusion injury (IRI) transgenic murine model, the administration or overexpression of IL-22 has been found to preserve renal function by increasing signal transducer and transcription factor 3 (STAT3) and protein kinase B (Akt) phosphorylation in proximal tubular epithelial cells, upregulating anti-apoptotic genes (e.g., *Bcl-2*), and downregulating pro-apoptotic genes (e.g., *Bad*) [63]. Retinoic acid (RA) signaling, activated in macrophages and tubular epithelial cells within hours of injury, has been found to reduce macrophage-dependent injury and fibrosis after AKI [64]. In zebrafish and murine models, the activation of RA signaling between epithelial cells and macrophages after AKI has been found to limit the injury extent by promoting the activation of M2 macrophages and tubular epithelial cell repair [64]. Wnt/β-catenin is another important pathway in recovery from AKI. Its activation by macrophages has been found to stimulate repair [65], and its early intervention is required to minimize renal damage after AKI in the initial phase [66]. However, its persistent activation and Wnt1 overexpression have been shown to play a role in progression to CKD through uncontrolled fibroblasts activation and inflammation [66,67]. Therefore, while important in facilitating repair after AKI, the presence of macrophages is also correlated with fibrosis and adverse outcomes. Moreover, the reciprocal expression of colony-stimulating factor-1 (CSF-1) and its receptor between macrophages and tubular epithelial cells could enhance cell proliferation and stimulate the anti-inflammatory M2 subtype [68]. Interestingly, to highlight the complexity of the interplay between these cells upon injury, IL-34 produced by injured TECs may have a pro-inflammatory ability despite sharing the same macrophage receptor of CSF-1 [69].

After the initial phase of injury, early inflammation is followed by the infiltration of circulating immune cells (T- and B-cells) attracted by cytokines and damage-associated molecular patterns (DAMPs) released by injured cells [27]. While interacting with activated monocytes/macrophages, injured TECs, and endothelial cells, DAMPs participate in the development of a pro-fibrotic milieu which activates pericytes to proliferate and evolve into myofibroblasts, thereby inducing matrix deposition, renal fibrosis, and CKD [70]. In contrast, a subset of regulatory T-cells (Treg) may act like self-tolerance inducers and suppress inflammation by enhancing immune homeostasis [71]. Together with their positive effects, Treg depletion has been shown to aggravate ischemic AKI [72]. Interestingly, the protective role of CD4+ and CD8+ T-cells in a murine model of toxic nephropathy (aristolochic acid nephropathy) has recently been described after treatment with selective monoclonal antibodies [73]. Concerning B-cells, immunoglobulins production, antigen presentation, and subsequent complement activation have been described as possible contributors to the progression from acute to chronic renal injury [74].

### 2.3. Fibrosis

Several obstructive, ischemia-reperfusion and nephrotoxic animal models have investigated the link between AKI and CKD by focusing on the development of interstitial fibrosis [31,32,33]. Myofibroblasts, whether derived from activated resident fibroblasts or from pericytes, are responsible for extracellular matrix (ECM) production, with the deposition of collagens, fibronectins and other glycoproteins, which, together with TGF-β, contribute to fibrosis [32,75]. The expression of α-smooth muscle actin (α-SMA), usually confined to the vascular compartment, and platelet-derived growth factor receptor-β (PDGFR-β), identifies these cells in the interstitium of injured kidneys [75]. A number of studies have consistently linked peritubular capillaries rarefaction, pericytes detachment, interstitial hypoxia and tubular epithelial injury as triggers of renal fibrosis [21,50]. In particular, whereas an ischemic injury could be responsible for capillary rarefaction and pericyte detachment, pericyte loss could be a trigger for endothelial damage and capillary rarefaction followed by tubular epithelial injury and fibrosis [50]. Though the severity of interstitial fibrosis in renal biopsies has been recognized as the major prognostic factor for CKD/ESRD, fibrosis has been considered a self-sustaining process [76], and a causal relationship between ECM deposition, fibrosis and chronic kidney injury has not yet been identified.

Recent studies have suggested that fibrosis could also be beneficial for the healing processes [34]. During repair from experimental AKI, tubules that fail to recover become atrophic, and fibrosis surrounds them in well-demarcated areas that separate the injured parenchyma from restored or not injured tubules [34]. In this view, fibrosis is itself essentially a self-limiting repair process that restricts injury, and it is not autonomously progressive. Indeed, many other experimental data do not support a major role of self-perpetuating tubulointerstitial fibrosis in the transition from AKI to CKD and highlight that progressive renal fibrosis requires additional injuries—unless primary interstitial disease is itself the triggering factor for fibrosis [23,77]. Recently, in a transgenic mouse model expressing diphtheria toxin receptor on renal fibroblasts, Nakamura et al. showed that fibroblasts depletion could worsen the expression of tubular injury markers, with a marked increase after unilateral ureteral obstruction [33]. While the transition of resident fibroblasts to myofibroblasts has been described to trigger fibrosis [78], myofibroblasts can also acquire retinoic acid-production ability—lost by the injured tubular epithelial cells—supporting epithelial integrity and repair [33] and dampening pro-inflammatory macrophages [64]. In the aged kidney, the ability of resident fibroblasts to support repair is less pronounced [79].

Altogether, a deeper understanding of the cellular and molecular pathways involving fibrosis in different types of acute kidney injury will be of great importance for the development of therapeutic strategies to halt the progression of AKI to CKD [80,81].

### 2.4. Tubular Epithelial Injury

The most sensitive cells to acute ischemic and nephrotoxic injury are the proximal S3 segment tubular epithelial cells of the outer stripe of the outer medulla due to their intense workload, high metabolic demand, and limited capacity for anaerobic energy production [82]. In fact, this region accounts for a unique microvascular environment which is extremely vulnerable to hypoperfusion, renal hypoxia, and mitochondrial damage [82,83]. Recently, a shift from a victim to the driving force of the AKI-to-CKD transition has been proposed for the tubular epithelial cell [35]. Indeed, injured TECs have been shown to act as drivers of both inflammation and fibrosis. They produce a large variety of cytokines (e.g., IL-6, IL-1β, and TNF-α), thereby gaining a pro-inflammatory phenotype and directly influencing macrophage behavior [84,85], and they are an important source of chemokines—via several pathways including STAT signaling and TGF-β signaling—adhesion molecules and reactive oxygen species [35]. Traditionally, injured TECs have been thought to undergo a process of dedifferentiation, a partial epithelial–mesenchymal transition (EMT) [86,87], i.e., the de novo expression of mesenchymal marker such as vimentin in TECs [88] induced by the injury-mediated reactivation of snail family zinc finger 1 (Snail1) [89,90]. This transient dedifferentiation is also characterized by the re-expression of developmental genes such as *Pax2* and cell cycle markers such as proliferating cell nuclear antigen (PCNA)—thus suggesting extensive proliferation—followed by a loss of mesenchymal markers, which has been interpreted as re-differentiation into fully viable epithelial cells [91,92]. This extensive proliferative ability could be responsible for rapid repair upon acute injury.

This proliferative capacity is thought to explain why young and mildly injured patients regain normal (or near-normal) renal function within days from acute injury, regardless of etiology. According to this view, the kidney’s reparative potential is counterbalanced by maladaptive repair, as if they are on a balance pan. Shifting from proliferation to maladaptive repair would explain disease progression. In one toxic and one obstructive nephropathy models, Yang et al. demonstrated a causal association between cell cycle arrest and fibrosis [93]. Due to abnormal repair processes, TECs can become atrophic and gain a pro-fibrotic phenotype after AKI. G2/M-arrested TECs may activate the JNK signaling to induce the production of pro-fibrotic cytokines (e.g., TGF-β and connective tissue growth factor (CTGF)) [22]. Altogether, whereas favorable cell cycle events could be responsible for repair, cell cycle arrest could determine the progression of injury. Dysregulated and inefficient (i.e., maladaptive) tubular repair has been related to the persistence of an inflammatory milieu, ECM deposition and subsequent tubular cells convergence towards a pro-fibrotic and senescent phenotype [94]. Indeed, dedifferentiated TECs acquire pro-fibrotic characteristics that elicit CKD progression [23]. 

An alternative pathophysiological interpretation of cell cycle events after acute injury has recently been proposed [95,96]. Indeed, several studies have recently pointed toward the existence of a scattered population of undifferentiated, self-renewing, renal progenitors with the ability to regenerate fully differentiated TECs rather than acquire a dedifferentiation state [97,98,99,100,101,102]. After their identification in the human kidney, further studies were able to provide detailed characterization in both humans and mice [100] and to identify these cells as a source of tubular regeneration after AKI [103]. This strategy for kidney regeneration appears to be highly conserved across species [104] and involved in kidney development, maintenance, and regeneration [105]. New experimental evidence provided by lineage tracing studies has strengthened the concept that tubular epithelial cell regeneration is mostly due to a scattered progenitors’ population rather than to the majority of remnant cells [95]. Renal progenitors are more numerous in the proximal tubule S3 segment, the segment which is more sensitive to ischemic and nephrotoxic injury and from where tubular cells detach in large numbers, thus explaining the high proliferation of tubular epithelial cells observed in this area (Figure 2) [95]. The intense immunoreactivity of nuclear proliferation markers (such as proliferating cell nuclear antigen-PCNA and Ki-67) has also been observed after acute injury in the proximal tubule S2 segment and other uninjured areas of the nephron [88,106]. Regardless, although such cell cycle markers confirm the cell’s entry in the cell cycle, they do not entail its completion with the formation of two new differentiated daughter cells. Thanks to simultaneous cell cycle phase lineage tracing analysis and DNA content measurement with FUCCI (fluorescent ubiquitination-based cell cycle indicator) technology, it has been shown that the majority of remnant TECs do enter the cell cycle, but they undergo endoreplication-mediated hypertrophy (Figure 2). Endoreplication is an evolutionary conserved cell cycle program by which cells replicate their genome without division, resulting in polyploid cells (i.e., polyploidization). Polyploidization increases the gene copy number in response to the need to quickly support increased functional requests for a higher metabolic output while persistently maintaining differentiated and specialized cell functions. This permits hypertrophy and function recovery [95,96].

In mammals, endoreplication-induced polyploidy has been observed in multiple tissue and organs (including the skin, placenta, liver, and blood) during normal development and under stressful conditions [107]. In the kidney, tubular cell polyploidization has frequently been observed in the proximal convoluted tubule S2 segment that is not directly injured during ATN (Figure 2). Accordingly, a new interpretation could follow: i) AKI causes TEC loss; ii) a small subset of progenitor cells showing resistance to death and proliferative ability are responsible for parenchymal regeneration; and iii) remnant TECs enter the cell cycle but undergo endoreplication-mediated polyploidy rather than mitosis, thus rapidly compensating for function loss [95].

According to this new hypothesis, the physiological response to AKI could imply a limited regeneration mediated by scattered renal progenitors and a polyploidization response by remnant TECs [96]. Polyploid TECs do not truly reconstitute parenchymal loss; thus, they might be a marker of irreversible loss and elicit progression towards chronic disease. In this view, a response to AKI is a costly process which cannot endlessly repeat without any consequence; rather, the tubular epithelium is more susceptible to further damage after every hit, better mirroring the clinical spectrum of the AKI-to-CKD transition.

## 3. Conclusions

A tight link between AKI and CKD is now becoming evident, both in the clinical and experimental settings. AKI severity, duration and frequency are associated with the development of CKD, but even mild episodes are associated with an increased risk of disease progression. Recent experimental findings have provided new insight into the cellular and molecular mechanisms of the AKI-to-CKD transition; these experiments have been trying to unveil the relative contribution of endothelial dysfunction, immune cell response, pericytes and fibroblasts activation. Regardless of the AKI etiology, endothelial dysfunction and subsequent hypoxia (as well as death of tubular cells from a toxic injury) trigger a cascade of self-sustaining events involving myofibroblast activation derived from resident fibroblasts or pericytes, extracellular matrix deposition, and interstitial inflammation. The overall view has shown that all the molecular and cellular mechanisms converge to the tubular epithelial cell dysfunction. Indeed, the lack of recovery of the tubular structure’s integrity sustains the above-mentioned events, thus promoting the progression of interstitial injury. The direct increase in the risk of CKD development and progression, which mirrors the severity of the acute episode, suggests a causative role of the final effector of acute function loss, i.e., epithelial cell injury. In several highly specialized organs, widespread parenchymal proliferation is likely to be a counterproductive strategy. Indeed, the mitosis and cytokinesis of highly specialized parenchymal cells determines a temporary loss of function that might become critical for the organ’s survival. It is now becoming evident that, to minimize the mitotic ability of specialized parenchymal cells without losing their functional performance, evolution has selected an alternative type of response: Hypertrophy via polyploidization. 

The biological rationale for increasing genome content through polyploidization could be to increase cell size and to facilitate amplified cell metabolism so that polyploid cells could sustain acute organ function recovery. Renal progenitor cells instead, thanks to their clonogenic ability, could be responsible for the true tissue regeneration and structural recovery of the necrotic S3 segment of the proximal tubule in affected nephrons, as highlighted by lineage tracing studies. Overall, these results suggest that injured tubules respond to ATN through two main mechanisms: The polyploidization of tubular cells and the proliferation of renal progenitors. New advances in the understanding of the biology and pathophysiology of epithelial tubular cells, renal progenitors, and their adaptation mechanisms will permit the better characterization of functional recovery after AKI and the tracing of the cascade of events leading towards CKD.

## Figures and Tables

**Figure 1 ijms-20-04941-f001:**
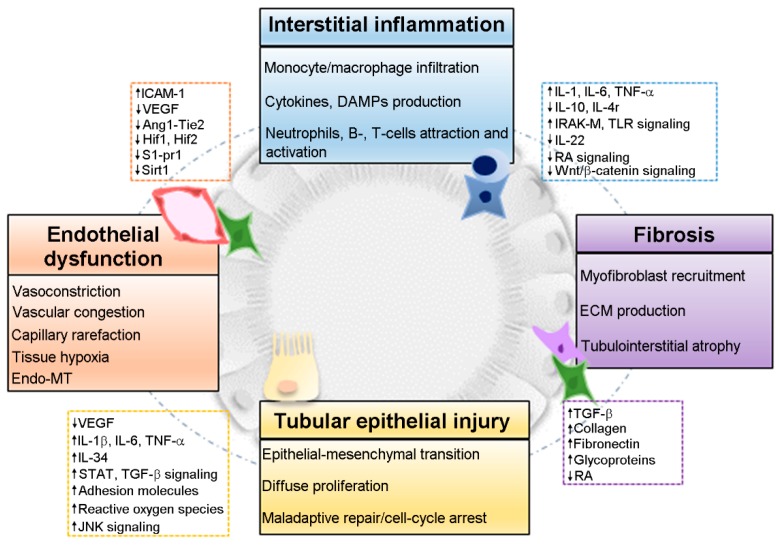
The interplay between endothelial dysfunction, interstitial inflammation, fibrosis and tubular epithelial injury concur to explain the acute kidney injury-to-chronic kidney disease (AKI-to-CKD) transition.

**Figure 2 ijms-20-04941-f002:**
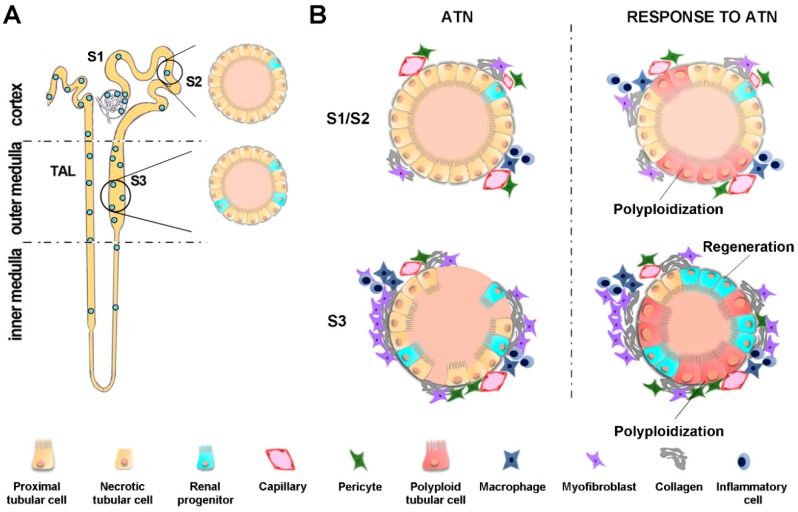
A new pathophysiological interpretation of tubular response to AKI leading towards CKD: The proliferation of renal progenitors and polyploidization of tubular cells. (**A**) Schematic localization of renal progenitors scattered along the S1–S2 segment, the S3 segment, and thick ascending limb (TAL) in the nephron. (Figure modified from Lazzeri et al., Trends Mol Med, 2019); (**B**) Top: In the uninjured proximal tubule S1–S2 segment, tubular epithelial cells enhance their working capacity by entering the cell cycle to increase their DNA content without division, resulting in polyploid tubular cells (i.e., polyploidization). Bottom: In the necrotic proximal tubule S3 segment, renal progenitors proliferate and complete cell division to drive regeneration, while the remnant tubular epithelial cells undergo polyploidization rather than mitosis. ATN: Acute tubular necrosis.

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
