# Peer review of "Molecular Mechanisms of the Acute Kidney Injury to Chronic Kidney Disease Transition: An Updated View"

_ijms, 2019, doi:10.3390/ijms20194941_

Round 1

Reviewer 1 Report

Molecular Mechanisms of AKI to CKD transition: an updated view.

By F. Guzzi e t al.

In this manuscript, the authors provide a brief review of mechanisms of transition from acute kidney injury to chronic kidney disease. They reviewed roles of endothelial injury, interstitial inflammation, fibrosis and tubular epithelial injury in the AKI to CKD transition. They underlined the major role of epithelial cells injury in the progression of CKD after primary acute insult.

This is an interesting and timely review that should be published. It is required only minor corrections.

Minor comments

Line 18. The sentence “The interplay between endothelial cells…” should be revised. It is unclear what type of the cells play central role in the AKI to CKD transition. Line 27. Section title need to be corrected. Line 43 “A recent systematic review be See at al…” Please provide a reference. Line 94 “Despite the existence…” Please correct the statement. It is too long and verb(s) is missing. Conclusion should be based on all sections, not only section 2.4.

Author Response

Point 1: Line 18. The sentence “The interplay between endothelial cells…” should be revised. It is unclear what type of the cells play central role in the AKI to CKD transition
Response 1: We revised the sentence according to the reviewer suggestion.
Point 2: Line 27. Section title need to be corrected.
Response 2: We corrected the title as suggested.
Point 3: Line 43. “A recent systematic review be See at al…” Please provide a reference.
Response 3: We provided the reference by See et al. as requested.
Point 4: Line 94. “Despite the existence…” Please correct the statement. It is too long and verb(s) is missing.
Response 4: We revised the sentence as suggested.
Point 5: Conclusion should be based on all sections, not only section 2.4.
Response 5: In agreement with the reviewer’s suggestion we expanded the conclusions speculating about all sections of the manuscript.

Reviewer 2 Report

Several reviews over the past five years, devoted to the progression of chronic kidney disease after acute kidney injury, have been published. The authors of the review paid special attention to a new pathophysiological hypothesis of tubular response to AKI leading towards CKD:  proliferation of renal progenitors and polyploidization of tubular cells. The review contains new data (2017-2019), illustrated by two figures.

The main comments:

Although the terms “AKI” and “CKD” are common, you should not use abbreviations in the title of the article. Make a list of abbreviations for easier review reading. In the References a very large proportion of links to reviews, compared with links to the original article.

Author Response

Point 1: Although the terms “AKI” and “CKD” are common, you should not use abbreviations in the title of the article.
Response 1: We modified the Title with the full forms of acute kidney injury and chronic kidney disease.
Point 2: Make a list of abbreviations for easier review reading.
Response 2: We added a list of abbreviations before the References section.
Point 3: In the References a very large proportion of links to reviews, compared with links to the original article.
Response 3: We reduced unnecessary review citations in the introductory section.